

# The role of geography and host abundance in the distribution of parasitoids of an alien pest

Petra Nováková[1], Jaroslav Holuša[2] and Jakub Horák[2]

[1] Department of Game Management and Wildlife Biology, Faculty of Forestry and Wood Sciences, Czech University of Life Sciences Prague, Prague, Czech Republic
[2] Department of Forest Protection and Entomology, Faculty of Forestry and Wood Sciences, Czech University of Life Sciences Prague, Prague, Czech Republic

## ABSTRACT

Chalcid wasps (Hymenoptera: Chalcidoidea) are probably the most effective and abundant parasitoids of the horse chestnut leaf miner (*Cameraria ohridella*), an alien pest in Europe that lacks specialized enemies. We studied how the species richness and abundance of chalcids are influenced by altitude, direction of an alien spread and host abundance of *C. ohridella*. We quantified the numbers and species richness of chalcid wasps and the numbers of *C. ohridella* that emerged from horse chestnut (*Aesculus hippocastanum*) leaf litter samples collected from 35 sites in the Czech Republic. Species richness of chalcids, which was considered an indicator of the possible adaptation of parasitoids to this alien host, was unrelated to *C. ohridella* abundance, direction of spread, or altitude. Chalcid abundance, which was considered an indicator of parasitism of the alien host, was strongly and positively related to *C. ohridella* abundance. Chalcid abundance was negatively related to direction of spread and positively related, although in a non-linear manner, to altitude. The relationship of chalcid abundance with direction of spread and altitude was weaker than that with *C. ohridella* abundance. The results provide evidence that biological control of the alien pest *C. ohridella* by natural enemies might develop in the future.

## INTRODUCTION

The horse chestnut leaf miner, *Cameraria ohridella* Deschka and Dimic, 1986 (Lepidoptera: Gracillariidae), is causing ecological problems throughout Europe (*Percival et al., 2011*; *Matosevic & Melika, 2012*). This species, which may have originated in the Balkans (*Valade et al., 2009*), has increased its distribution (*Sefrova & Lastuvka, 2001*) within a relatively short time (*Augustin, 2013*). Although *C. ohridella* was not described until 1986, DNA analysis of herbarium specimens indicates that the species was present in Europe at least as early as 1879 (*Lees et al., 2011*).

In addition to causing aesthetic damage, mining by *C. ohridella* larvae may weaken or even kill horse chestnut trees (*Aesculus hippocastanum* L.)—the mining is nearly

Corresponding author
Jakub Horák, jakub.sruby@gmail.com

constant throughout the growing season because the insect has multiple, overlapping generations (*Matosevic & Melika, 2012*). The weakened trees increase the dustiness in urban environments and reduce the food supply for game in non-urban environments (*Percival et al., 2011*). This alien pest also harms native fauna (*Pere et al., 2010*) and other tree species in Europe (*Freise, Heitland & Sturm, 2004*).

Because *C. ohridella* overwinters as pupa in leaves that have fallen to the ground, *C. ohridella* numbers can be reduced by leaf removal (*Gilbert et al., 2003*; *Kehrli & Bacher, 2003*). Leaf removal, however, is time consuming and thus expensive. In addition, the removed leaves must be properly composted to prevent leaf miner emergence in the following spring (*Kehrli & Bacher, 2004*). Burning is not always possible because of weather or local regulations. *C. ohridella* may also be controlled by the use of insecticides or pheromones but these methods have been inconsistent in reducing the abundance of this pest and may harm native fauna (*Wagner et al., 1996*; *Sefrova, 2001*). Although the application of synthetic inhibitors of chitin synthesis proved to be very effective (*Blumel & Hausdorf, 1997*; *Percival, Banks & Keary, 2012*), the residues of these inhibitors may be highly stable (i.e., persistent) on horse chestnut leaves (*Nejmanova et al., 2006*). From a long-term perspective, breeding of horse chestnut tree with resistance to *C. ohridella* is an option (*Mertelik, Kloudova & Vanc, 2004*).

The current research concerns the control of *C. ohridella* by natural enemies. Among the approximately 60 generalist parasitoids of *C. ohridella* (e.g., *Grabenweger & Lethmayer, 1999*; *Toth & Lukas, 2005*), Chalcid wasps (Hymenoptera: Chalcidoidea) are considered the most abundant and effective control agents (*Grabenweger & Lethmayer, 1999*). The overall parasitism rate of non-native *C. ohridella* by indigenous enemies is affected by temporal factors (i.e., miner residence time) and spatial factors (i.e., geography) (*Grabenweger et al., 2010*). In addition, the attack of alien pests by native natural enemies is often delayed—as a consequence, the alien pest often suffers little biological control early in its invasion (*Godfray, 1994*; *Schonrogge & Crawley, 2000*).

Many geographical factors influence the spread, expansion and distribution of organisms, and especially important predictors are altitude (*Lomolino et al., 2010*) or direction of spread (*Sefrova & Lastuvka, 2001*). These factors are often correlated with climate. Altitude, as an example, is known to well reflect geographical heterogeneity (*Tognelli & Kelt, 2004*). At the spatial scale of the Czech Republic, south-to-north expansion of *C. ohridella* was correlated with latitude of *C. ohridella* (*Sefrova & Lastuvka, 2001*). The relative importance of geography and host distribution on the distribution of parasitoids depends on the host-specificity of the parasitoids, i.e., the effect of host distribution becomes more important as host-specificity increases (*Sivinski, Pinero & Aluja, 2000*; *Skillen, Pickering & Sharkey, 2000*).

To our knowledge, *C. ohridella* lacks host-specific natural enemies (*Grabenweger & Lethmayer, 1999*; *Toth & Lukas, 2005*). Thus, we suspect that the number of species and individuals of non-specific parasitoids may be able to successfully respond to the high abundance of this alien pest only if the parasitoids are limited by geography only marginally.

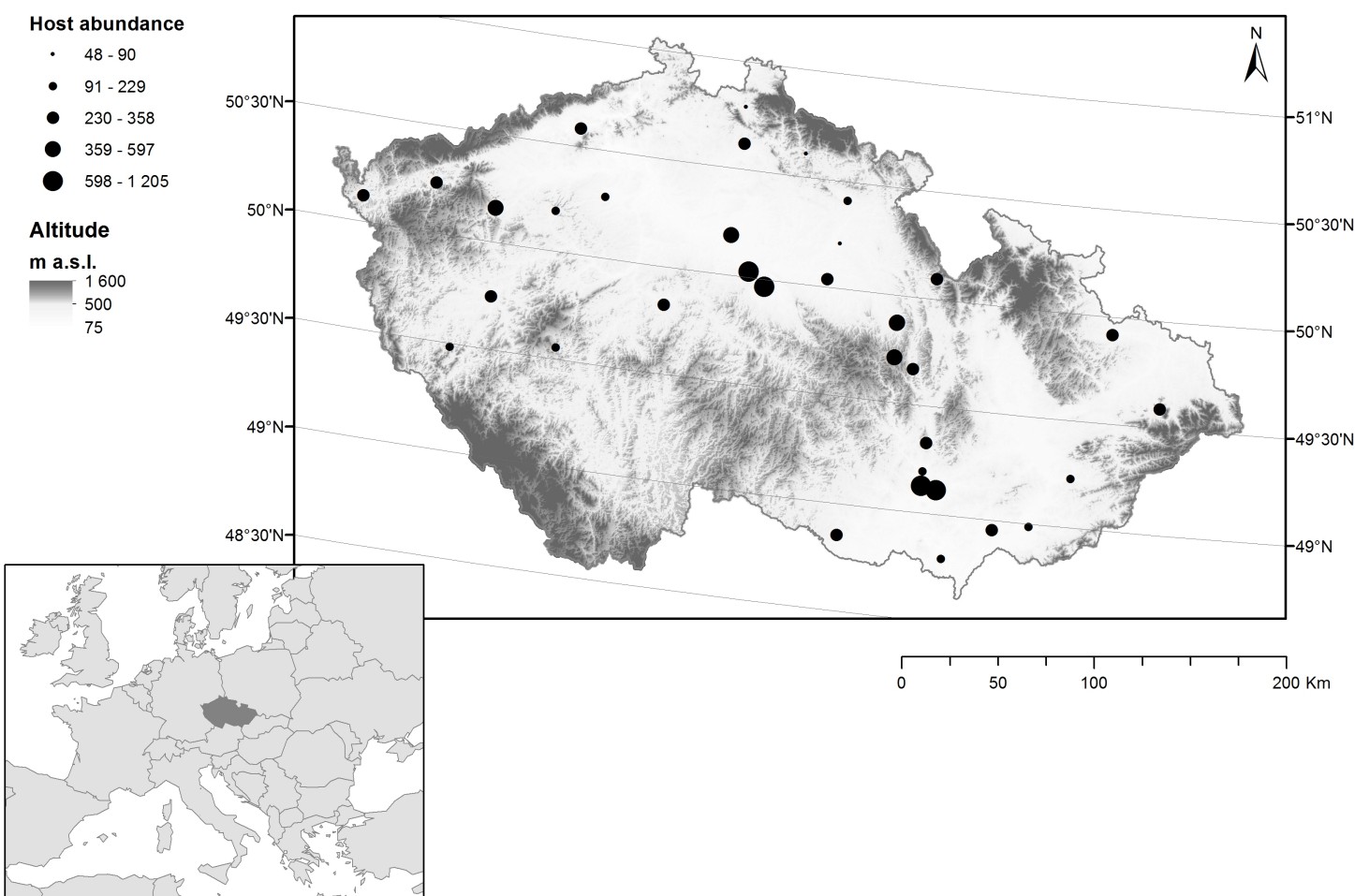

**Figure 1** **Distribution of sampling sites containing horse chestnut trees (*Aesculus hippocastanum*) infested with the horse chestnut leaf miner (*Cameraria ohridella*) in the Czech Republic.** The abundance of *C. ohridella* (based on data collected in the current study) is indicated by black circles, altitude is indicated by grey shading, and latitude is indicated by grey lines.

The main aim of this study was the answer on the question: How are the species richness and abundance of chalcid parasitoids influenced by geography and by the abundance of the alien pest, *C. ohridella*?

## MATERIAL AND METHODS

### Sampling sites

During 2002, we studied the parasitism of the horse chestnut leaf miner by chalcid wasps in 35 sites in the Czech Republic (Fig. 1), Each site contained a road lined with ≥ five horse chestnut trees that were infested with *C. ohridella*. These sites are typical of the patches with horse chestnut trees in the Czech Republic and are known to be highly suitable for *C. ohridella* development (*Sefrova & Lastuvka, 2001*).

### Study methods

Horse chestnut leaf litter samples were collected from the soil surface under the crowns of horse chestnut trees that were distant from other tree species to minimize the possibility
**Table 1** Variance inflation factors (VIF), means, standard errors, and ranges of the three studied predictors: host abundance (the number of *Cameraria ohridella* adults that emerged from each litter sample), site altitude (m a.s.l.), and direction of spread (°).

| Predictor | VIF | Mean | SE | Minimum | Maximum |
|---|---|---|---|---|---|
| Host abundance | 1.1 | 354.6 | 48.5 | 48.0 | 1,205.0 |
| Altitude | 1.2 | 319.1 | 20.5 | 173.0 | 575.0 |
| Spread direction | 1.1 | 49.8 | 0.1 | 48.8 | 50.8 |

that the litter was contaminated with leaves of other species. All samples were taken during the early spring before the emergence of parasitoids (*Grabenweger, 2004*). At each site, we collected 1 m² (≈0.1 m height) of pure horse chestnut leaf litter.

All samples were covered with paper sheets and immediately transported to the laboratory, where the litter was placed in emergence traps (cardboard boxes 0.6 × 0.9 × 0.2 m) at 18–20°. All arthropods that emerged from the litter were trapped in 70% ethanol. The adult *C. ohridella* and chalcids were counted daily. The ethanol was replaced daily and the preserved chalcids were identified to species.

## Dependent variables and environmental predictors

Dependent variables included the number of chalcid species and the number of individuals of chalcid wasps that emerged from each litter sample. We used the list of *Nováková & Nakladal (2008)* for preliminary comparison of the parasitoid species and we found that all reared species are known to be associated with *C. ohridella*. It is indicated that species in their native areas are hosts of a higher number of species of parasitoids (*Girardoz, Kenis & Quicke, 2006*; *Grabenweger et al., 2010*). Thus, the number of chalcid species was considered an indicator of possible adaptation of parasitoids to the alien host, i.e., an increase in species would suggest an increase in adaptation. On the other hand, the number of chalcid individuals was considered a possible indicator of chalcid abundance and rate of parasitism of the alien host (e.g., *Arneberg et al., 1998*).

We studied three environmental predictors (Table 1) that might influence the species richness and abundance of parasitoids that emerge from litter samples. The number of *C. ohridella* adults that emerge (Host abundance) reflects *C. ohridella* abundance at each site. Altitude of the site reflects geographical heterogeneity and correlates with climate. The direction of spread (Spread direction) might well reflect the situation of species richness and abundance of parasitoids during the time of active spread of invasive species. Thus, the direction of spread was used as the third environmental predictor. As the spread of *C. ohridella* in the Czech Republic had south-to-north direction (*Sefrova & Lastuvka, 2001*; *Sefrova, 2003*), degrees of latitude were used.

## Statistical analyses

All analyses were conducted in R 3.0.2 (*R Development Core Team, 2013*). The potential bias caused by spatial autocorrelation was controlled by Moran's correlograms using the spdep package (*Bivand, 2005*). Because our data did not show spatial bias at any distance ($I < -0.1$; $P > 0.1$), we used traditional statistical methods.

**Table 2** Relationships between the number of species and abundance of chalcid wasps (Hymenoptera: Chalcidoidea) that emerged from litter samples infested with *Cameraria ohridella* and predictors (Host abundance, Altitude and Spread direction) as indicated by hierarchical partitioning and generalized linear models.

| Dependent variable | AIC | Predictor | TEV | z | P |
|---|---|---|---|---|---|
| Number of chalcid species per site | 107.85 | Host abundance | 13.3 | 1.0 | n.s. |
| | | Altitude | 12.3 | −1.0 | n.s. |
| | | Spread direction | 2.2 | −0.1 | n.s. |
| Number of chalcid individuals per site | 541.33 | Host abundance | 48.9 | 17.2 | <0.001 |
| | | Altitude | 1.9 | 4.2 | <0.001 |
| | | Spread direction | 7.7 | −4.5 | <0.001 |

**Notes.**
AIC, Akaike Information Criterion; TEV, % of total explained variance.

We then controlled for possible circular predicting and multicolinearity using the HH package (*Heiberger, 2009*) and the value of variance inflation factor (VIF). This showed that *C. ohridella* abundance was not correlated with the other studied predictors (Table 1), i.e., with altitude ($R = -0.3$; $P = $ n.s.) or latitude ($R = -0.3$; $P = $ n.s.). Data for the number of species and individuals of chalcids had Poisson distributions.

The variance explained by the predictors was computed using $R^2$ in hierarchical partitioning (package hier.part; *Walsh & Mac Nally, 2011*). The relationships between the dependent variables and the predictors were computed using generalized linear models and generalized additive models with the gam package (*Hastie, 2011*). Generalized additive models were fitted by spline function.

## RESULTS

A total of 811 individuals (mean = 23.2 ± 4.3 SE; min = 1; max = 118) of eight chalcid wasp species (1.9 ± 0.2; 1–4) emerged from the 35 litter samples, namely: *Cirrospilus viticola* (0.1 ± 0.1), *Closterocerus trifasciatus* (0.8 ± 0.4), *Pediobius saulius* (2.1 ± 1.4), *Pnigalio agraules* (11.5 ± 2.9), *Pnigalio pectinicornis* (0.7 ± 0.3), *Pteromalus semotus* (1.4 ± 0.6) *Minotetrastichus frontalis* (7.1 ± 1.6) and *Sympiesis sericeicornis* (1 individual).

The number of parasitoid species that emerged was not significantly related to the studied predictors (Table 2). The number of chalcid individuals that emerged (i.e., chalcid abundance) was positively related to the number of *C. ohridella* that emerged from each sample, i.e., *C. ohridella* abundance explained nearly 50% of the variance in chalcid abundance. Chalcid abundance was negatively related with spread direction, and spread direction explained nearly 8% of the variance in chalcid abundance (Table 2). Unexpectedly, chalcid abundance was positively related with altitude (Table 2), although the response to altitude was not linear (Fig. 2 and Table 3). Altitude explained less than 2% of the variance in chalcid abundance (Table 2).

## DISCUSSION

Data in the current study were collected when the invasive horse chestnut leaf miner (*C. ohridella*) had become fully established in the Czech Republic (*Sefrova & Lastuvka,*

**Table 3** Relationships between the number of chalcid wasps (Hymenoptera: Chalcidoidea) that emerged from litter samples infested with *Cameraria ohridella* and predictors (Host abundance, Altitude and Spread direction) as indicated by the generalized additive model (DF = 1.5).

| Name | AIC | Deviance | Predictor | Npar $\chi^2$ | P |
|---|---|---|---|---|---|
| Number of chalcid individuals per site | 495.71 | 56.02% | Host abundance | 5.5 | <0.01 |
| | | | Altitude | 31.9 | <0.001 |
| | | | Spread direction | 10.1 | <0.001 |

**Notes.**
AIC, Akaike Information Criterion; Npar $\chi^2$, non-parametric value of $\chi^2$.

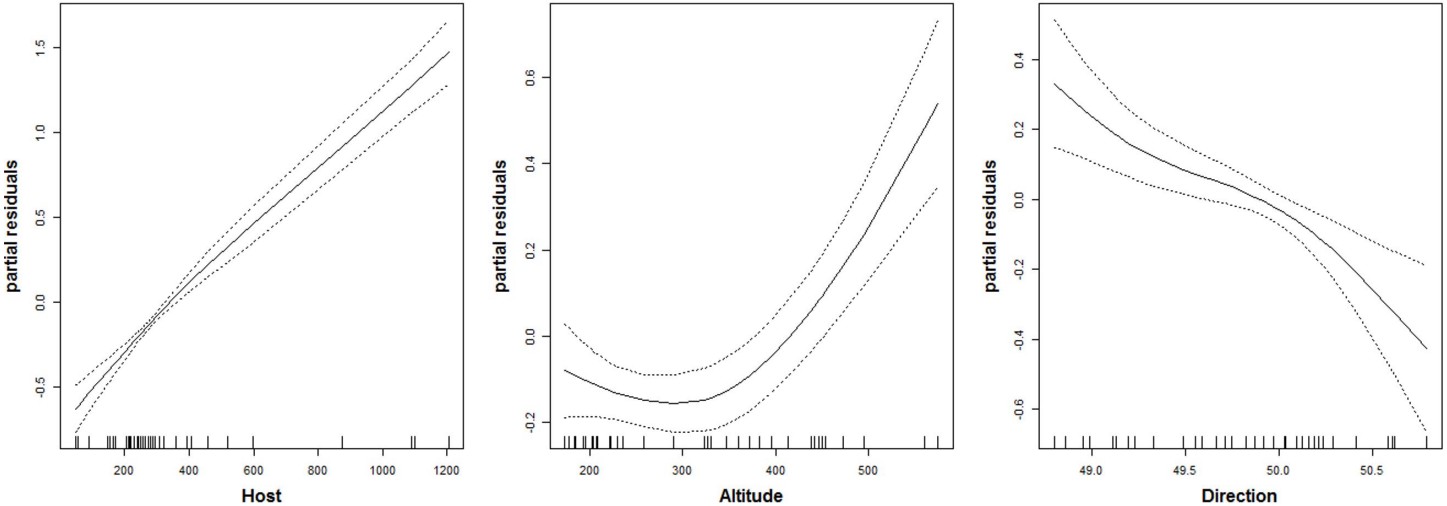

**Figure 2** **Relationship between the abundance of chalcid wasps (Hymenoptera: Chalcidoidea) and three predictors.** Host is abundance of *C. ohridella*; Altitude is m a.s.l.; and Direction is spread direction from north-to-south as indicated by the generalized additive model fitted by spline function with DF = 1.5.

*2001*). Since then, this invasive pest has expanded throughout central Europe and has established its first populations on the British Isles (*Augustin, 2013*). Our results show that the abundance of parasitoids of *C. ohridella* was weakly related to predictors that are highly connected with geography, namely altitude and spread direction (linked to the latitude), but was relatively strongly related to *C. ohridella* abundance.

The number of parasitoid species was not significantly related to the studied predictors. This result indicates that the adaptation of indigenous parasitoid species to the alien pest was rather low, which is consistent with *Girardoz, Kenis & Quicke (2006)* and it seemed that most of the parasitoid species were generalists, which agrees with *Novakova & Nakladal (2008)*. On the other hand, parasitoid abundance was closely and positively related to *C. ohridella* abundance.

In addition to being closely related to *C. ohridella* abundance, the abundance of generalist parasitoids seemed relatively high, even though the emergence of *C. ohridella* and its parasitoids are indicated to be poorly synchronized (*Grabenweger, 2004*). Although parasitism rates as high as 50% have been reported for other leaf mining moths, the percentage of *C. ohridella* attacked by parasitoids is often low and does not usually reach

20% (*Grabenweger & Lethmayer, 1999*; *Novakova & Nakladal, 2008*; *Grabenweger et al., 2010*). This low parasitism rate, which undoubtedly contributed to the heavy infestation of horse chestnut trees by *C. ohridella* in many places, probably results from former insufficient adaptation of the local parasitoids to this recently introduced leaf miner. If such adaptation is possible, it will most probably require more time (*Zwölfer & Pschorn-Walcher, 1968*).

On the other hand, we suspect that generalist parasitoids may adapt to *C. ohridella* given that their abundance increased with that of the pest although with delay. *Grabenweger et al. (2010)* hypothesized that the adjustment of specialist parasitoids requires more than a few decades. Recruitment and accumulation of native parasitoid species on introduced herbivores has been documented (*Cornell & Hawkins, 1993*), and exotic insects do not necessarily suffer lower enemy-induced mortality rates than natives (*Hawkins, Cornell & Hochberg, 1997*). A quick shift of native parasitoids to the new invasive host *Tuta absoluta* (Meyrick 1917) was observed in Italy (*Zappala et al., 2012*). Similarly, another recent study indicated that resident generalist parasitoids and predators can work in conjunction to hinder the invasion of a herbivore (*Hogg et al., 2013*). It follows that although natural enemies have not prevented invasion of Europe by *C. ohridella*, based on our results we could suppose that successful biological control of invasive *C. ohridella* by natural enemies may develop in the future—because the total amount of chalcid individuals can better reflect the rate of parasitism of the alien host than number of adapted parasitic species.

The relationship to the spread direction fairly well illustrated that the number of parasitoids is decreasing with increasing distance from the area of origin. On the other hand, the increasing number of individuals of parasitoids was higher in higher altitudes, which is not common (*Lomolino et al., 2010*). This might correlate with relatively high altitude of the Lake Ohrid and surrounding areas in Macedonia and Albania, which is the area of origin of *C. ohridella* (*Valade et al., 2009*). The result appears to indicate that aliens are more vulnerable to enemies in conditions that are close to their former area of distribution (e.g., *Roy et al., 2011*).

## CONCLUSIONS

The number of parasitoid species that emerged from leaf litter infested with the horse chestnut leaf miner, *C. ohridella*, was not significantly related to *C. ohridella* abundance, altitude or spread direction, a finding which possibly indicates a delayed response of indigenous enemies to the expansion of their hosts. Although the abundance of generalist parasitoids was weakly related to altitude and spread direction, it was strongly related to *C. ohridella* abundance. Our results indicate a potential for biological control of *C. ohridella* by generalist parasitoids.

## ACKNOWLEDGEMENTS

We dedicate this paper to the memory of Dr. Zdeněk Bouček, who helped us with the determination of chalcid wasps. The authors thank Dr. Bruce Jaffee for linguistic improvements and Petr Vopěnka for the map.

### Funding

The research was supported by project NAZV KUS QJ1520197 of the Ministry of Agriculture and partly supported by the Internal Grant Agency (IGA no. A18/15), Faculty of Forestry and Wood Sciences, Czech University of Life Sciences Prague. The funders had no role in study design, data collection and analysis, decision to publish, or preparation of the manuscript.

### Grant Disclosures

The following grant information was disclosed by the authors:
NAZV KUS: QJ1520197.
IGA FFWS CULS Prague: A18/15.

### Competing Interests

The authors declare there are no competing interests.

### Author Contributions

- Petra Nováková conceived and designed the experiments, performed the experiments, contributed reagents/materials/analysis tools.
- Jaroslav Holuša contributed reagents/materials/analysis tools, wrote the paper, reviewed drafts of the paper.
- Jakub Horák analyzed the data, contributed reagents/materials/analysis tools, wrote the paper, prepared figures and/or tables, reviewed drafts of the paper.

### Data Availability

The raw data has been supplied as Data S1.

### Supplemental Information

Supplemental information for this article can be found online at http://dx.doi.org/10.7717/peerj.1592#supplemental-information.

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
