# Peer review of "The role of geography and host abundance in the distribution of parasitoids of an alien pest"

_PeerJ, doi:10.7717/peerj.1592_

## Round 0.1 · original submission · Major Revisions

Reviewer #1 (who reviewed this MS in its previous iteration) finds the paper mainly acceptable now with some grammatical errors that need fixing, etc.

Reviewer #2 is a new reviewer with a fresh set of eyes. They did have access to the previous reviews and my previous comments. They make a number of points that would improve the paper substantially, and so I am sticking with their recommendation as my recommendation in this case. I will ask Reviewer #2 if they would be fine with reviewing the revised version of this MS once the authors have done the required work and/or provided rebuttal.

Thank you to the three reviewers (on this and the previous iteration) and the authors for your work through this process.

Reviewer 1 ·

Basic reporting

This is the second review I provide for this article. I have read the cover letter explaining the revisions and I have also read the word documents in track changes, especially the points that I raised in the first review.
I am satisfied that the authors have included the methodological details missing; particularly the information on the species collected and their mean abundance. Also they have focused their discussion more towards their results rather than having a very general discussion of local parasitoid attacking alien invaders.
I notice that the paper still contains some small grammar errors that the authors should fix prior to publication.

Experimental design

nothing new to add to my first review

Validity of the findings

nothing new to add to my first review

Additional comments

nothing new, except to correct some small grammar errors

Reviewer 2 ·

Basic reporting

See "General Comments for the Author"

Experimental design

See "General Comments for the Author"

Validity of the findings

See "General Comments for the Author"

Additional comments

The authors appear to mostly have addressed the concerns from previous reviews, but I think a few issues remain. Ultimately I think it will make a useful contribution to the literature but there is still some work to do - briefly I would say more data and potentially analysis in the Results section, and setting the Discussion in broader context by discussing other possible causes of the patterns seen in more detail.

Firstly a minor point, which is that I think the title is a little misleading – given the approach used I’m not sure you can really say that you are disentangling geography and host abundance, I think you’d really need to use an experimental approach to tease those apart. A more appropriate title could be possibly created from some rewording of the question posed in lines 83-85. This is more of a suggestion though.

Secondly, I’m in agreement with some of the previous comments by the editor and reviewers, and I’m not entirely clear on the response to the previous review – are there any data at all (even at order level) on other arthropods that were collected? I only ask because in my own experiences rearing parasitoids from field collected materials, any spiders or other predators that are collected can sometimes wipe out anything else in rearing containers. Might just help with explaining some of the results.

Thirdly and possibly most importantly, I think more discussion of alternative causes for the patterns is still required to set the broader context of the manuscript. This was brought up in the previous reviews and I don’t think it has been sufficiently addressed. There are clearly myriad possible factors affecting the distribution of these parasitoids, and the authors have to be really careful about their conclusions, as looking at three different factors (even if some do potentially represent multiple environmental variables) is probably insufficient. There are still so many other possible causes for the parasitoid distributions, e.g., evolutionary history, precipitation, temperature variation/extremes, interspecific competition, land use, presence of alternate hosts (given the apparent generalist nature of many of the parasitoids) to name but a few, and it would be helpful if there was more discussion on these.

This leads into another point, which is that I think the authors overstate the implications of their results somewhat. I’m not convinced that the results at present suggest that biological control is likely to work for the leaf miner. There is very little presented to suggest that the parasitoids are responding to the leaf miner, and no data on the host abundance, so there’s no way to tell if the parasitoids have any effect on population growth. That’s not to say of course that it might not occur in future, but based on what’s presented in the paper, I don't think you can say that there is substantial potential for biological control. Some of the statements made by the authors also contradict this idea that there is good potential for biological control (e.g., lines 180-184). Additionally, it would be really helpful to know more about the timeline and spread of the leaf miner invasion in Europe and more specifically the Czech Republic. At the moment some of the statements are quite vague and it’s a little hard to grasp how, where and when it invaded.

Some other more minor comments/suggestions below.

Introduction:
L71: remove ‘an’

Results:
It would be useful if the authors provided some kind of abundance or relative abundance for the wasp species they found. This could just be in the form of a table of with numbers in parentheses next to each species in the text, or something along those lines. At the moment it just seems like the Results section is quite light when there are a lot of data that could be presented. Knowing the relative abundances could also help to inform the Discussion. Admittedly, I would not consider myself an expert on European chalcids, but it would be useful to know more about the biology and ecology of the wasp species actually collected, e.g., is much known about how host specific they are? Or that they all attack and can complete a life cycle C. ohridella? Other common hosts? There is a very brief mention that ‘most of the parasitoids were generalists’ on line 177 (and a little in the introduction), but, for instance, what is known about the most abundant parasitoid species found?

Also, did the authors consider using some kind of diversity index in addition to richness and abundance separately? Again just wondering if it might help to explain trends in the results.

Discussion
L168: should be ‘data in the current study’

Figures:
L342: remove one ‘in’, and change ‘indicate’ to ‘indicated’

---

## Round 0.2 · Minor Revisions

The reviewer finds the MS mainly acceptable, with some very minor revisions.

Thanks to the reviewers of the various iterations of this MS, and to the authors for working along with the recommendations. Following minor revisions, this MS should be acceptable for publication in PeerJ.

Reviewer 2 ·

Basic reporting

See General Comments for the Author

Experimental design

See General Comments for the Author

Validity of the findings

See General Comments for the Author

Additional comments

The authors have done a thorough job addressing my comments from my previous review, and I have no further concerns with the manuscript. Just a few minor things below.

I think the new title is better, but maybe ‘The role of geography and host abundance in the distribution of parasitoids of an alien pest’ would better reflect the study?

L77: change to ‘becomes’
L114: change to ‘a higher number of’
L200: remove ‘but’
L218: I would prefer to see ‘substantial’ removed or replaced with a more cautious term. I agree that there is potential for biocontrol, but as I mentioned in my previous review I don’t really think the present study explicitly demonstrates that in a strong way (not a criticism of the study, just more of a limitation based on what you were looking at).

---

## Round 0.3 · accepted · Accept

Thank you for your work on this MS. The MS is now acceptable for publication in PeerJ.